Genome-wide analysis of sugar transporter genes in maize (Zea mays L.): identification, characterization and their expression profiles during kernel development

Sun Nan 1 2
Liu Yanfeng 1 2
Xu Tao 1 3
Zhou Xiaoyan 1 3
Xu Heyang 1 3
Zhang Hongxia 1 2
Zhan Renhui 4
Wang Limin wanglimin9696@163.com 1 2
1 The Engineering Research Institute of Agriculture and Forestry, Ludong University , Yantai , Shandong , China
2 Zhaoyuan Shenghui Agricultural Technology Development Co., Ltd. , Zhaoyuan , Shandong , China
3 College of Agriculture, Ludong University , Yantai , Shandong , China
4 School of Pharmacy, Shandong Technology Innovation Center of Molecular Targeting and Intelligent Diagnosis and Treatment, Binzhou Medical University , Yantai , Shandong , China
Mansour Elsayed
Electronic publication date: 2023 Nov 17
Publication date: 2023
Volume: 11
Electronic Location ID: e16423
Received 2023 Jul 31; Accepted 2023 Oct 18
Copyright: ©2023 Sun et al.
Copyright year: 2023
Copyright holder: Sun et al.
License: This is an open access article distributed under the terms of the Creative Commons Attribution License, which permits unrestricted use, distribution, reproduction and adaptation in any medium and for any purpose provided that it is properly attributed. For attribution, the original author(s), title, publication source (PeerJ) and either DOI or URL of the article must be cited.
License URL: https://creativecommons.org/licenses/by/4.0/

Keywords: Zea mays, Sugar transporters, Phylogenetic analysis, Gene expression, Gene family

Funding: The Natural Science Foundation of Shandong Province, China ZR2020QC115 ZR2019PC015 The Cooperation Project of University and Local Enterprise in Yantai of Shandong Province 2021XDRHXMPT09 Modern Agricultural Industry Technology System Innovation Team of Shandong Province of China SDAIT-02-05 Double-Hundred Talents Project of Yantai City This work has been jointly supported by the following grants: The Natural Science Foundation of Shandong Province, China (ZR2020QC115, ZR2019PC015); The National Natural Science Foundation of China (31901572, 32071733); The Cooperation Project of University and Local Enterprise in Yantai of Shandong Province (2021XDRHXMPT09); the Modern Agricultural Industry Technology System Innovation Team of Shandong Province of China (SDAIT-02-05) and the Double-Hundred Talents Project of Yantai City. The funders had no role in study design, data collection and analysis, decision to publish, or preparation of the manuscript.

==============================
Sugar transporters (STs) play a crucial role in the development of maize kernels. However, very limited information about STs in maize is known. In this study, sixty-eight ZmST genes were identified from the maize genome and classified into eight major groups based on phylogenetic relationship. Gene structure analysis revealed that members within the same group shared similar exon numbers. Synteny analysis indicated that ZmSTs underwent 15 segmental duplication events under purifying selection. Three-dimensional structure of ZmSTs demonstrated the formation of a compact helix bundle composed of 8–13 trans-membrane domains. Various development-related cis-acting elements, enriched in promoter regions, were correlated with the transcriptional response of ZmSTs during kernel development. Transcriptional expression profiles exhibited expression diversity of various ZmST genes in roots, stems, leaves, tassels, cobs, embryos, endosperms and seeds tissues. During kernel development, the expression of 24 ZmST genes was significantly upregulated in the early stage of grain filling. This upregulation coincided with the sharply increased grain-filling rate observed in the early stage. Overall, our findings shed light on the characteristics of ZmST genes in maize and provide a foundation for further functional studies.

Introduction

In higher plants, sugars, including monosaccharide and sucrose, play a crucial role in enhancing yield (Büttner, 2007; Julius et al., 2017). In many plant species, sucrose is synthesized in green organs (source) and transported over long distances through the phloem to heterotrophic organs (sink) (Van Bel, 2003). Upon reaching the sink organs, sucrose is either directly transported into sink cells or cleaved into monosaccharides by cell wall-bound invertases, which are subsequently taken up by the sink cells (Sherson et al., 2003). Extensive researches have established that the transport of sugars into sink cells is mediated by sugar transporters (STs), which facilitate the transport of both monosaccharides and sucrose (Noiraud, Maurousset & Lemoine, 2001; Büttner, 2007; Kühn & Grof, 2010).

Many sugar transporters, specifically those from major facilitator superfamily (MFS) and sugar will eventually be exported transporters (SWEET) family, have been identified in various species (Chen et al., 2012; Zheng et al., 2014). MFS is further divided into the monosaccharide transporter (MST) family and the sucrose transporter (SUT) family, with MST family exhibiting greater diversity (Yan, 2013). The MST family members are classified into seven subfamilies, including sugar transporter proteins (STPs) that act as proton/sugar symporters for various monosaccharides (Büttner, 2007), polyol/monosaccharide transporters (PMTs) responsible for transporting monosaccharide and sugar alcohols on the plasma membrane (Noiraud, Maurousset & Lemoine, 2001), sugar facilitator proteins (SFPs) that export hexoses on the vacuolar membrane (Yamada et al., 2010; Klemens et al., 2014), inositol transporters (INTs) that function as H+/inositol symporters (Strobl et al., 2018), plastidic glucose translocators (pGlcTs) that export glucose into the cytosol (Cho et al., 2011), and two families of monosaccharide importers for sugar uptake in the tonoplast, namely tonoplast sugar transporters (TSTs) and vacuolar glucose transporters (VGTs) (Aluri & Büttner, 2007; Cheng et al., 2018a; Cheng et al., 2018b). These eight families of MFS-type sugar transporters are ancient and present in both dicotyledonous and monocotyledonous plants (Lemoine, 2000; Johnson, Hill & Thomas, 2006). The SWEET family, discovered in 2010, belongs to another superfamily and possesses seven transmembrane domains (Chen et al., 2010; Xuan et al., 2013). Due to these differences, the SWEET transporter family will not be discussed in this study.

Previous studies have shown the importance of STs in the transportation of sugars to sink tissues, which is crucial for crop yield and quality. In Arabidopsis, there are 62 identified AtST genes. Mutants of AtSUC2, which have decreased sucrose transport in the phloem, accumulate excessive starch in the leaves, leading to severe growth inhibition and reduced fertility (Gottwald et al., 2000; Gould et al., 2012). The expression of AtSTP4 gradually increases during pollen development, with the highest level occurring in mature pollen (Truernit et al., 1996). AtSTPs are not been found in the female gametophyte or developing seeds (Büttner, 2010). AtVGT1, located on vacuolar membrane, plays an important role in flowering and seed germination by transporting glucose (Aluri & Büttner, 2007). AtTMT1 and AtTMT2 transport monosaccharides and sucrose into the vacuole (Schulz et al., 2011). Mutants of Aterdl6 show increased vacuolar glucose levels and increased seed weight due to higher sugar, protein, and lipid levels (Poschet et al., 2011).

In rice, OsTMTs transport glucose into vacuoles and contribute to sugar storage in vacuoles (Cho et al., 2010). OsSUT1 is involved in long-distance sucrose transport, plant height, pollen vitality and seed germination. Mutants of OsSUT1 exhibit a slight dwarf phenotype and complete infertility due to failed grain filling (Hu et al., 2021; Sun et al., 2022; Wang et al., 2022). OspGlcT2 is expressed in response to sugar and salt, indicating its role in salt stress tolerance (Deng et al., 2019). OsSTP10 is induced by sucrose and fructose treatments in roots, but does not respond to hormone treatments. OsSTP16 is highly expressed in flag leaf sheaths and responds rapidly to glucose and fructose (Deng et al., 2019). The OsTMTs in rice function similarly to AtTMTs, transporting monosaccharides into vacuoles (Cho et al., 2010).

Maize (Zea mays L.) is a significant global food crop with important economic and social value, as well as applications in the bioenergy industries (Tian et al., 2019). Additionally, it serves as an excellent model organism for genetic and genomic studies due to its high photosynthetic rate, availability of a reference genome and efficient transformation system (Schnable et al., 2009; Wang et al., 2020). STs not only play a role in sugar transport and allocation, but also have crucial impacts on plant yield and quality. However, compared to species like strawberry, pear, tomato and rice, limited research has been conducted on the ST gene family in maize. In this study, we performed a comprehensive search against the updated maize genome B73_RefGen_v5 and identified 68 ZmST genes. Through phylogenetic relationship, chromosome location, collinearity analysis, conservative structures and expression patterns analyses in maize, we found that ZmSTs play a significant role in sugar transportation and seed development. These results serve as valuable references for further research on ZmSTs and provide new genetic resources for the high-yield maize breeding.

Materials & Methods

Plant materials and growth condition

The maize inbred line B73 was used in this study. B73 seeds were sterilized with mercuric chloride and cultured in ddH2O at 28 °C (light)/23 °C (dark) with a 16 h light/8 h dark photoperiod (Li et al., 2013). After germination, the seedlings with uniformed growth were selected and moved into the field. Subsequently, various tissues and kernels at different days after pollination were collected for the analysis of ZmST expression levels.

Identification and characterization of ST proteins in maize

To investigate putative ST genes in Zea mays, two methods were employed. First, 62 AtST sequences in Arabidopsis were downloaded and used to perform a BLAST search against the Zea mays genome obtained from maizeGDB (https://www.maizegdb.org/) with default parameters (Long et al., 2021). Additionally, the Hidden Markov Model (HMM) profiles of the Sugar_tr domain (PF00083), MFS-1 (PF07690) and MFS-2 (PF13347) were obtained from Pfam (http://pfam.xfam.org/) and utilized for HMMER 3.0 searches against the potential ST proteins in maizeGDB (Prakash et al., 2017; Mistry et al., 2021). All potential ZmST proteins were determined on NCBI (https://www.ncbi.nlm.nih.gov/cdd/) and SMART (https://smart.embl.de/).

Comparison of the numbers of ST gene families in different plants

ST genes from various plants, including Arabidopsis (Büttner, 2007), rice (Deng et al., 2019), tomato (Reuscher et al., 2014), pear (Li et al., 2015), strawberry (Liu et al., 2020), grape (Afoufa-Bastien et al., 2010), Longan (Fang et al., 2020), and apple (Wei et al., 2014) were analyzed to compare the number of STs across different plant species.

Chromosomal location, collinearity and duplication event analyses

The chromosomal locations of ZmST genes on chromosomes and chromosome synteny were performed by TBtools (Chen et al., 2020). Gene duplication analyses were conducted as previously described. The ratio of non-synonymous substitution rate (Ka) to synonymous substitution rate (Ks) was calculated by TBtools.

Phylogenetic tree analysis of STs from different plants

Amino acid sequences of STs from Zea mays, Arabidopsis thaliana and Oryza sativa were used to create a phylogenetic tree. The phylogenetic tree was constructed by MEGA 7.0 software using neighbor-joining (NJ) phylogenetic method with 1,000 bootstrap replications (Kumar, Stecher & Tamura, 2016).

Gene structure, conserved motif and domain analyses

The gene structure of ZmST genes was analyzed by TBtools software. The conserved motifs of ZmST proteins were analyzed with MEME (https://meme-suite.org/meme/) (Bailey et al., 2009). The maximum number of predicted motifs was set to 15. The final graph was presented by TBtools.

Expression heatmap of transcriptome

RNA-Seq datasets from different tissues were acquired from maizeGDB to analyze the expression profiles of the ZmST genes (Stelpflug et al., 2015). Ten tissues from maize vegetative development to reproductive development stages were used to identify tissue specificity of ZmST genes. The expression data of STs was visualized using the TBtools.

RNA extraction and qRT-PCR

RNAs from B73 materials were extracted by RNAprep Pure Plant Kit (Tiangen Biotech Co., Ltd., Beijing, China) according to the manufacturer’s instruction. About 1-2 µg of RNA was using to reverse transcribe with HiScript® III All-in-one RT SuperMix reverse kit reagents (Vazyme Biotech Co., Ltd.). In order to eraser gDNA and synthesize cDNA, 4 µl of 5 × All-in-one RT SuperMix, 1 µl of Enzyme Mix, 1–2 µg of RNA and appropriate volume of RNase-free ddH2O were mixed. Subsequently, the mixture was incubated at 50 °C for 15 min, followed by a temperature increase to 85 °C for 5 s. The qRT-PCR was performed with primers listed in Table S1, with ZmACTIN1 as an internal reference. qPCR was run on CFX96TM real-time system (Bio-Rad, Hercules, CA, USA), with ChamQ Universal SYBR qPCR Master Mix (Vazyme Biotech Co., Ltd., Nanjing, China), as previously described (Fang et al., 2023). Finally, the calculation method for ZmST genes expression levels was proposed by Livak and Schmittgen (Livak & Schmittgen, 2001).

Cis-acting regulatory elements analysis in the ZmST gene promoters

The promoter regions of ZmSTs were obtained with TBtools software for promoter analysis. The cis-acting regulatory elements were identified by PlantCARE (http://bioinformatics.psb.ugent.be/webtools/plantcare/html/) and presented with TBtools.

Results

Sixty-eight ZmST genes are identified in maize genome

In this study, a total of 68 gene sequences encoding putative ST proteins were identified. Their physicochemical properties, including gene ID, protein size, molecular weight (MW), isoelectric point (p I), the grand averages of hydropathicity (GRAVY), and localization prediction, were characterized (Table 1). The molecular weight of ZmST proteins ranged from 41.83 kDa (ZmSFP1) to 80.96 kDa (ZmTST3), while the isoelectric points ranged from 4.72 (ZmTST3) to 9.82 (ZmSTP12) (Table 1). The grand averages of hydropathicity for all ST proteins indicated their hydrophobic nature. Subcellular localization analysis revealed that all ZmST proteins were located in the cell membrane (Table 1, Table S2).

Table 1 Physicochemical characteristics of 68 ST proteins.

Gene name	Gene ID	Protein size (aa)	MW a (KDa)	p I b	GRAVY c	Localization prediction	
ZmINT1	Zm00001eb425560	509	53.92	5.31	0.609	Cell membrane	
ZmINT2	Zm00001eb300060	585	62.44	8.82	0.361	Cell membrane	
ZmINT3	Zm00001eb306230	500	53.18	9.06	0.482	Cell membrane	
ZmINT4	Zm00001eb426370	591	63.84	8.67	0.364	Cell membrane	
ZmpGlcT1	Zm00001eb125210	539	56.81	9.22	0.567	Cell membrane	
ZmpGlcT2	Zm00001eb335350	539	56.59	9.08	0.585	Cell membrane	
ZmpGlcT3	Zm00001eb311910	550	58.08	6.27	0.466	Cell membrane	
ZmpGlcT4	Zm00001eb203690	485	52.35	8.59	0.535	Cell membrane	
ZmPMT1	Zm00001eb008070	556	59.39	8.13	0.335	Cell membrane	
ZmPMT2	Zm00001eb027550	508	54.03	9.22	0.541	Cell membrane	
ZmPMT3	Zm00001eb066030	562	58.31	7.14	0.579	Cell membrane	
ZmPMT4	Zm00001eb075840	534	58.04	6.01	0.431	Cell membrane	
ZmPMT5	Zm00001eb008080	524	56.15	9.08	0.444	Cell membrane	
ZmPMT6	Zm00001eb325680	509	54	8.88	0.617	Cell membrane	
ZmPMT7	Zm00001eb021140	519	55.3	9.53	0.558	Cell membrane	
ZmPMT8	Zm00001eb107810	516	54.37	9.16	0.621	Cell membrane	
ZmPMT9	Zm00001eb325640	522	55.59	8.9	0.61	Cell membrane	
ZmPMT10	Zm00001eb107870	520	55.08	9.16	0.589	Cell membrane	
ZmPMT11	Zm00001eb325650	513	54.26	9	0.612	Cell membrane	
ZmPMT12	Zm00001eb411020	489	50.58	8.7	0.599	Cell membrane	
ZmPMT13	Zm00001eb411000	478	50.16	8.75	0.742	Cell membrane	
ZmPMT14	Zm00001eb166230	501	52.14	9.25	0.633	Cell membrane	
ZmPMT15	Zm00001eb166250	487	50.94	8.89	0.658	Cell membrane	
ZmPMT16	Zm00001eb166210	481	50.2	8.79	0.73	Cell membrane	
ZmSFP1	Zm00001eb017730	386	41.83	9.03	0.69	Cell membrane	
ZmSFP2	Zm00001eb017760	510	54.33	6.91	0.547	Cell membrane	
ZmSFP3	Zm00001eb127290	492	52.06	8.33	0.644	Cell membrane	
ZmSFP4	Zm00001eb333940	485	51.43	5.67	0.634	Cell membrane	
ZmSFP5	Zm00001eb344570	506	54.09	9.23	0.635	Cell membrane	
ZmSFP6	Zm00001eb296190	500	53.74	8.54	0.552	Cell membrane	
ZmSFP7	Zm00001eb296190	502	54.16	8.31	0.58	Cell membrane	
ZmSFP8	Zm00001eb296220	642	68.5	9.25	0.476	Cell membrane	
ZmSFP9	Zm00001eb344010	499	53.28	8.49	0.58	Cell membrane	
ZmSFP10	Zm00001eb344020	496	52.62	9.08	0.632	Cell membrane	
ZmSFP11	Zm00001eb344040	499	52.92	9.24	0.613	Cell membrane	
ZmSTP1	Zm00001eb298310	523	57.04	9.38	0.467	Cell membrane	
ZmSTP2	Zm00001eb008810	514	56.69	9.05	0.507	Cell membrane	
ZmSTP3	Zm00001eb000240	525	57.5	9.21	0.515	Cell membrane	
ZmSTP4	Zm00001eb324180	524	56.91	9.13	0.606	Cell membrane	
ZmSTP5	Zm00001eb182870	521	56.35	9.21	0.609	Cell membrane	
ZmSTP6	Zm00001eb171880	514	55.43	7.54	0.647	Cell membrane	
ZmSTP7	Zm00001eb391140	520	56.83	9.37	0.503	Cell membrane	
ZmSTP8	Zm00001eb043150	536	56.85	9.35	0.601	Cell membrane	
ZmSTP9	Zm00001eb207680	521	56.91	9.1	0.603	Cell membrane	
ZmSTP10	Zm00001eb159990	509	55.65	9.54	0.672	Cell membrane	
ZmSTP11	Zm00001eb110690	510	55.64	9.43	0.686	Cell membrane	
ZmSTP12	Zm00001eb377440	518	54.52	9.82	0.581	Cell membrane	
ZmSTP13	Zm00001eb047630	508	54.32	9.14	0.554	Cell membrane	
ZmSTP14	Zm00001eb309780	522	57.27	9.41	0.47	Cell membrane	
ZmSTP15	Zm00001eb098100	518	56.63	9.16	0.527	Cell membrane	
ZmSTP16	Zm00001eb312640	523	56.2	9.34	0.557	Cell membrane	
ZmSTP17	Zm00001eb303000	513	56.14	9.04	0.483	Cell membrane	
ZmSTP18	Zm00001eb244790	522	55.82	9.58	0.58	Cell membrane	
ZmSTP19	Zm00001eb423910	456	49.86	9.71	0.627	Cell membrane	
ZmSTP20	Zm00001eb081130	513	54.77	9.12	0.609	Cell membrane	
ZmTST1	Zm00001eb022230	747	79.55	4.85	0.401	Cell membrane	
ZmTST2	Zm00001eb239520	745	79.82	5.26	0.397	Cell membrane	
ZmTST3	Zm00001eb228740	763	80.96	4.72	0.316	Cell membrane	
ZmTST4	Zm00001eb166700	652	71.7	5.64	0.354	Cell membrane	
ZmVGT1	Zm00001eb225000	518	55.46	5.72	0.598	Cell membrane	
ZmVGT2	Zm00001eb211520	559	58.71	9.6	0.624	Cell membrane	
ZmSUT1	Zm00001eb005460	521	55.17	8.58	0.608	Cell membrane	
ZmSUT2	Zm00001eb133930	501	53.37	8.84	0.486	Cell membrane	
ZmSUT3	Zm00001eb048470	508	53.52	7.46	0.584	Cell membrane	
ZmSUT4	Zm00001eb259340	592	63.14	6.63	0.323	Cell membrane	
ZmSUT5	Zm00001eb244930	530	56.2	8.63	0.494	Cell membrane	
ZmSUT6	Zm00001eb183000	530	55.94	8.7	0.554	Cell membrane	
ZmSUT7	Zm00001eb402200	519	55.09	8.68	0.609	Cell membrane	
Notes.

a MW, molecular weight.

b pI, isoelectric point.

c GRAVY, grand averages of hydropathicity.

ZmST proteins are divided into eight groups

A neighbor-joining tree with 199 STs, including 68 ZmSTs from maize, 62 AtSTs from Arabidopsis, and 69 OsSTs from rice, was constructed (Fig. 1). The phylogenetic tree suggested that the sugar transporters in maize were classified into eight groups. Among them, the VGT clade consisted of ZmVGT1 and ZmVGT2, while the STP clade contained ZmSTP1 to ZmSTP20. Additionally, the PMT, SFP, SUT, INT, pGlcT and TST clades included 16, 11, 7, 4, 4, and 4 members, respectively, and were annotated as ZmPMT1 to ZmPMT16, ZmSFP1 to ZmSFP11, ZmSUT1 to ZmSUT7, ZmINT1 to ZmINT4, ZmpGlcT1 to ZmpGlcT4, ZmTST1 to ZmTST4 (Fig. 1). Furthermore, phylogenetic analysis showed that there were some closely related orthologous STs between maize and rice, implying the existence of a set of ancestral ST genes before the divergence of the two species (Fig. 1).

Figure 1 Phylogenetic analysis of sugar transporters from Z. mays, A. thaliana and O. sativa.

A total number of 68 ZmSTs from maize, 62 AtSTs from Arabidopsis and 69 OsSTs from rice were used to construct the phylogenetic tree by MEGA 7.0 using the neighbor-joining (NJ) method with 1,000 bootstrap replications. All sugar transporter members were classified into eight groups (STP, PMT, SFP, SUT, TST, pGlcT, INT, VGT). The red triangle, yellow square and green circle signs represented Z. mays, A. thaliana and O. sativa, respectively.

Additionally, a comparison of the numbers of different ST groups among maize, Arabidopsis, rice, tomato, pear, woodland strawberry, grape, longan, and apple was carried out (Table 2). The results revealed that STP and PMT were the largest clades in maize, consistent with previous findings in rice, pear, and apple. Conversely, in Arabidopsis, tomato, strawberry, grape, and longan, the largest clades were STP and SFP.

Table 2 Comparative analysis the gene numbers of different ST families in maize, Arabidopsis, rice, tomato, pear, strawberry, grape, longan and apple.

Subfamily	Number of genes	
	Maize	Arabidopsis	Rice	Tomato	Pear	Strawberry	Grape	Longan	Apple	
STP	20	14	28	18	20	24	22	20	30	
PMT	16	6	15	8	23	7	5	6	10	
SFP	11	19	6	10	5	16	22	10	8	
SUT	7	9	5	3	6	8	4	6	9	
INT	4	4	3	4	6	3	3	4	4	
pGlcT	4	4	4	4	6	3	4	3	4	
TST	4	3	6	3	6	3	3	1	5	
VGT	2	3	2	2	3	2	2	2	3	
Total	68	62	69	52	75	66	65	52	73	

Figure 2 Chromosomal distribution and collinearity analysis of ZmST genes.

(A) Chromosome distribution of ZmSTs in maize genome using TBtools software. The chromosomal location of each ZmST gene was mapped according to the maize genome. The chromosome number is indicated at the top of each chromosome. (B) ZmST gene duplications analysis with TBtools. The syntenic ZmST gene pairs are connected by red lines.

Segmental duplication events are observed in ZmSTs

To investigate features of the ZmSTs gene family, we analyzed the chromosome distribution of each ZmST gene. Our findings revealed that the ZmST genes were located on all 10 chromosomes of maize (Fig. 2A). Chromosome 1 had the highest number of ZmST genes, including ZmPMT1, ZmPMT2, ZmPMT5, ZmPMT7, ZmSFP1, ZmSFP2, ZmSTP2, ZmSTP3, ZmSTP8, ZmSTP13, ZmTST1, ZmSUT1, and ZmSUT3. Chromosome 2 contained ZmPMT3, ZmPMT4, ZmPMT8, ZmPMT10, ZmSTP11, ZmSTP15, and ZmSTP20. ZmpGlcT1, ZmSFP3, ZmSTP10, and ZmSUT2 were located on chromosome 3. Chromosome 4 harbored ZmpGlcT4, ZmPMT14, ZmPMT15, ZmPMT16, ZmSTP5, ZmSTP6, ZmSTP9, ZmTST4, and ZmSUT6. Chromosome 5 contained ZmSTP18, ZmTST2, ZmTST3, ZmVGT1, ZmVGT2, ZmSUT4, and ZmSUT5. Chromosome 6 had ZmSFP6, ZmSFP7, and ZmSFP8 genes, while chromosome 7 contained ZmINT2, ZmINT3, ZmpGlcT3, ZmPMT6, ZmPMT9, ZmPMT11, ZmSTP1, ZmSTP4, ZmSTP14, ZmSTP16, and ZmSTP17. Chromosome 8 harbored ZmpGlcT2, ZmSFP4, ZmSFP5, ZmSFP9, ZmSFP10, and ZmSFP11. Chromosome 9 contained ZmSTP7, ZmSTP12, and ZmSU7, while chromosome 10 harbored ZmINT1, ZmINT4, ZmPMT12, ZmPMT13, and ZmSTP19 (Fig. 2A). Notably, all members of the ZmVGT group were located on chromosome 5, and four members of the ZmINT group were evenly distributed on chromosomes 7 and 10. Chromosome 6 only contained three ZmSFP members. Aside from these observations, the distribution of other ZmST genes on maize chromosomes was uneven.

Gene duplications are important for the expansion of gene families (Cannon et al., 2004; Konrad et al., 2011). Collinear analysis showed that 15 gene pairs, ZmpGlcT1 and ZmpGlcT2, ZmPMT7 and ZmPMT9, ZmPMT7 and ZmPMT10, ZmPMT8 and ZmPMT9, ZmSFP3 and ZmSFP4, ZmSFP6 and ZmSFP9, ZmSTP5 and ZmSTP18, ZmSTP5 and ZmSTP19, ZmSTP5 and ZmSTP20, ZmSTP18 and ZmSTP19, ZmSTP18 and ZmSTP20, ZmSTP19 and ZmSTP20, ZmSUT1 and ZmSUT3, ZmSUT1 and ZmSUT7, ZmSUT3 and ZmSUT7, have undergone segmental duplication events (Fig. 2B). Further analysis revealed that all the Ka/Ks values of the ST gene pairs were less than 1, indicating that the duplication events occurred under purifying selection (Fig. 2B, Table 3).

ZmST gene structures are highly conserved

The gene structures play crucial roles in the evolution and functional diversification of multiple gene families (Lei et al., 2020). The structural analysis of ZmSTs indicated that all ZmST genes, except ZmSTP6, harbored at least one intron. Several genes, namely ZmSFP1, ZmSFP2, ZmSFP3, ZmSFP4, ZmSFP5, ZmSFP6, ZmSFP7, ZmSFP8, ZmSFP9, ZmSFP10, ZmSFP11, ZmpGlcT1, ZmpGlcT2, ZmpGlcT3, ZmpGlcT4, ZmVGT1, ZmVGT2, ZmSUT1, ZmSUT4 and ZmSUT7 contained more than ten introns. However, genes within the same group usually had a similar number of exons (Figs. 3A and 3B). ZmSFP1 gene had 19 exons, whereas ZmSTP6 had only one exon, which indicated that the exons gain and loss occurred during the evolution of ZmST gene family. The structures of ZmSTs in duplication pairs, such as ZmpGlcT1 and ZmpGlcT2, ZmPMT7 and ZmPMT10, ZmSFP3 and ZmSFP4, ZmSTP5 and ZmSTP19, ZmSTP5 and ZmSTP20, ZmSUT1 and ZmSUT7, were highly similar.

We used the MEME online tool to predict the potentially conserved motifs of 68 ZmSTs. Among the 15 distinct motifs identified, only motif 6 was present in all ZmST proteins (Figs. 3C and 3D, Table S3). Notably, significant differences were observed in the conserved motifs between the SUT subfamily and MST subfamilies, despite their similar function in sugar transport (Figs. 3C and 3D). Motifs 1, 2, 4 and 5 were present in all 61 MST proteins, but were absent in SUT proteins, while motifs 13 and 15 existed in all 7 SUT proteins but not in MST proteins. These findings indicated that motifs 1, 2, 4 and 5 may be critical for the function of MST subfamilies, while motifs 13 and 15 may be necessary for the function of SUT subfamily. This may be due to functional differences between MST subfamilies (monosaccharide transport) and SUT subfamily (sucrose transport). Although all MST proteins contained motifs 1, 2, 4 and 5, the specific types and numbers of motifs varied among each subfamily. Motif 3 was absent in VGT subfamily and ZmSTP3 protein. Motif 7 was not present in any members of the SFP subfamily but was observed in the VGT subfamily. Motif 12 was absent only in ZmINT3 and ZmpGlcT4. Motif 14 was only observed in STP subfamily. Similar motif structures were observed in gene pairs such as ZmpGlcT1 and ZmpGlcT2, ZmPMT7 and ZmPMT9, ZmPMT7 and ZmPMT10, ZmPMT8 and ZmPMT9, ZmSFP3 and ZmSFP4, ZmSFP6 and ZmSFP9, ZmSTP5 and ZmSTP18, ZmSTP5 and ZmSTP20, ZmSTP18 and ZmSTP20, as well as ZmSUT1 and ZmSUT7.

Table 3 The Ka/Ks for the duplication gene pairs in ZmST family.

Duplicated pair	Duplicate type	Ka	Ks	Ka/Ks	Positive selection	
ZmpGlcT1/ZmpGlcT2	Segmental	0.02302462	0.17639281	0.13053037	No	
ZmPMT7/ZmPMT9	Segmental	0.1887097	0.73309884	0.25741372	No	
ZmPMT7/ZmPMT10	Segmental	0.18220538	0.59309584	0.30721068	No	
ZmPMT8/ZmPMT9	Segmental	0.05922607	0.19262997	0.30746033	No	
ZmSFP3/ZmSFP4	Segmental	0.12261777	0.72591107	0.16891569	No	
ZmSFP6/ZmSFP9	Segmental	0.14390645	0.54307605	0.26498398	No	
ZmSTP5/ZmSTP18	Segmental	0.21542769	0.59259227	0.36353442	No	
ZmSTP5/ZmSTP19	Segmental	0.35185515	0.66335247	0.5304196	No	
ZmSTP5/ZmSTP20	Segmental	0.37342149	0.55823017	0.66893821	No	
ZmSTP18/ZmSTP19	Segmental	0.37160006	0.71051129	0.52300373	No	
ZmSTP18/ZmSTP20	Segmental	0.35668412	0.69017173	0.51680489	No	
ZmSTP19/ZmSTP20	Segmental	0.1612672	0.44168081	0.3651216	No	
ZmSUT3/ZmSUT7	Segmental	0.22699235	0.76180805	0.29796528	No	

Figure 3 Exon-intron structure and conserved motifs of ZmST gene family.

(A) Phylogenetic tree of 68 sugar transporters in maize. The phylogenetic tree was constructed by MEGA 7.0 with 1,000 bootstrap replications. Bootstrap values above 50% were considered significant and indicated on the branch nodes. (B) Gene structure analysis of ZmST genes. Yellow blocks, black lines and green blocks represented exons, introns and untranslated regions, respectively. (C) The conserved motifs in ZmSTs. The different colored boxes represented different motifs. (D) Sequence logos for 15 conserved motifs were performed using MEME online tool. The x-axis represented the width of the motif and the y-axis represented the bits of each letter.

ZmST proteins were predicted to form a compact helix bundle

In order to explore the potential roles of ZmST proteins, we predicted their conserved domains and 3D models of all ZmSTs with NCBI-CDD and Swiss-model, respectively. All ZmST proteins contained a conserved MFS domain, which facilitated the transportation of various substrates (including sugars, ions, nucleosides, amino acids and so on) through the cytoplasm or inner membrane, except SUT clade. This SUT clade comprised a conserved GPH_sucrose superfamily domain, which might export sucrose from photosynthetic sources to the phloem or import sucrose into sucrose sinks (Fig. 4). Furthermore, 3D prediction demonstrated that all the ZmSTs were folded into 8–13 transmembrane domains, and then formed a compact helix bundle (Figs. 5, S1). However, it’s worth noting that most members within the same group exhibited a similar 3D structure. For instance, the ZmTST subfamily members displayed a unique central loop, composed of approximately 320 amino acids, which connected to the predicted transmembrane domains, a feature absent in all other sugar transporters (Fig. 5).

Figure 4 Analysis of the conserved domains in ZmST proteins.

(A) Phylogenetic tree of 68 sugar transporters in maize. (B) The conserved domains in ZmSTs were identified with NCBI-CDD. The conserved domains were presented with different colors.

Figure 5 Predicted 3D structures of ZmST proteins.

Different subfamilies were represented by different colors. All ZmSTs were folded into 8–13 transmembrane domains to form a compact helix bundle.

ZmST promoters contain the cis-acting elements for light, phytohormone, stress and development

The regulatory cis-elements are the binding sites for transcription factors, carrying information to regulate the gene expression in biological pathways. Thus, we extracted the promoter regions of 68 ZmST genes and examined their cis-acting elements using PlantCARE database 5.0. A total of sixty-six cis-elements were identified and categorized into four main groups: photoresponse, hormonal response, stress response and development. Among these cis-elements, thirty were related to light-responsive pathway, indicating their role in responding to light, which aligned with the function of sugar transporters in the distribution of photosynthetic products. Additionally, fourteen were associated with hormone response, eleven with abiotic and biotic stress response, and eleven with plant growth and development (Fig. 6). All promoters of ZmSTs had the same number of CGTCA-motif and TGACG-motif, and most ZmSTs were regulated by abscisic acid and methyl-jasmonate. ZmSUT4 and ZmSFP9 had seven and nine CCGTCC-boxes in their promoters, respectively, indicating their potential involvement in meristem-specific activation. The GCN4_motif was present in the promoters of ZmINT2, ZmPMT8, ZmPMT16, ZmSFP2, ZmSFP6, ZmSFP8, ZmSTP4, ZmSTP6, ZmSTP9, ZmSTP17, ZmTST3, ZmSUT4 and ZmSUT5, suggesting their potential role in endosperm expression. The RY-element was found in the promoters of ZmINT2, ZmPMT8, ZmPMT16, ZmSFP2, ZmSFP6, ZmSFP8, ZmSTP4, ZmSTP6, ZmSTP9, ZmSTP17, ZmTST3, ZmSUT5 and ZmSUT7, indicating their potential involvement in seed-specific regulation. The most abundant cis-elements in ZmST promoter regions were G-box, ABRE and STRE, implying that ZmSTs may participate in maize’s growth, development and response to light, hormones and stress.

Figure 6 The cis-acting elements in the promoters of ZmST gene family.

The names of 66 cis-elements were labeled at the bottom of the figure.

ZmSTs exhibit distinctive expression profiles across ten different tissues in maize

We downloaded the transcriptome data for ZmST genes and generated an expression pattern map using data from ten different tissues: young leaves, mature leaves, old leaves, roots, stems, tassels, cobs, embryos, endosperms, and seeds. Our analysis revealed that nineteen ZmST genes (ZmINT1, ZmpGlcT1, ZmpGlcT2, ZmpGlcT3, ZmpGlcT4, ZmPMT4, ZmPMT13, ZmSFP4, ZmSFP5, ZmSFP7, ZmSFP9, ZmSTP16, ZmTST1, ZmTST2, ZmTST4, ZmVGT1, ZmVGT2, ZmSUT2 and ZmSUT4) showed constitutive expression (log2 (FPKM+1) ≥ 1). Among these genes, ZmpGlcT1, ZmpGlcT2, ZmpGlcT4, ZmSFP5, ZmTST1, ZmTST2, ZmTST4, ZmSUT2 and ZmSUT4 showed the highest expression levels across all tested tissues and organs (log2(FPKM+1) ≥ 3) (Fig. 7). On the other hand, eleven ZmST genes (ZmINT3, ZmPMT14, ZmPMT15, ZmSFP2, ZmSFP3, ZmSTP6, ZmSTP9, ZmSTP17, ZmSTP18, ZmSTP19 and ZmSUT3) showed no expression across all tested tissues and organs (Fig. 7). Furthermore, we observed that two ZmST genes (ZmPMT1 and ZmPMT7), four genes (ZmPMT3, ZmPMT6, ZmSTP5, and ZmSTP14), and four genes (ZmSTP10, ZmSTP11, ZmTST3, and ZmSUT6) exhibited specific expression only in old leaves, roots, and tassels, respectively, with hardly detected expression in other organs. Additionally, we found that ZmSTP and ZmPMT groups displayed high expression in vegetative organs, whereas ZmPMT group members were scarcely detected in developing seeds. However, ZmTST group members showed higher transcription level in the developing seeds (Fig. 7).

Figure 7 Expression patterns of ZmSTs in 10 tissues.

The genes were labeled on the right and the tissues were displayed at the bottom of each column. YL, young leaf when the ninth leaf is fully unfolded; ML, mature leaf when the ninth leaf is fully unfolded; OL, old leaf in blister stage; Ro, crown roots node5 when the seventh leaf is fully unfolded; St, stem when the third leaf is fully unfolded; Tassel, miotic tassel when the eighteenth leaf is fully unfolded; Co, immature cob when the eighteenth leaf is fully unfolded; Em16, Em18, Em20, Em22, Em24, Em38: embryo of 16 DAP (days after pollination), 18 DAP, 20 DAP, 22 DAP, 24 DAP, 38 DAP, respectively; En12, En14, En16, En18, En20, En22, En 24: endosperm of 12 DAP, 14 DAP, 16 DAP,18 DAP, 20 DAP, 22 DAP, 24 DAP, respectively; Se2, Se4, Se6, Se8, Se10, Se12, Se14, Se16, Se18, Se20, Se22, Se24: whole seed of 2 DAP, 4 DAP, 6 DAP, 8 DAP, 10 DAP, 12 DAP, 14 DAP, 16 DAP, 18 DAP, 20 DAP, 22 DAP, 24 DAP, respectively.

Most sugar transporter genes showed a notable upregulation during the early stage of grain filling

As shown in Fig. 7, the diverse expression patterns of ZmST genes during stages of embryo, endosperm, and seed development suggested a significant role of ZmSTs in maize kernel development. To explore the probable functions of ZmST genes, we randomly selected 24 ZmST genes representing eight groups and executed qRT-PCR analysis across embryo and endosperm developmental stages as well as seed maturity stages. ZmpGlcT2, ZmSFP5, ZmSTP3, ZmTST1, ZmTST2, ZmVGT1, ZmVGT2 and ZmSUT2 showed up-regulation during embryo development and down-regulation during endosperm development. Conversely, ZmSTP7 were down-regulated during embryo development but up-regulated during endosperm development. Some genes like ZmINT4, ZmPMT4, ZmPMT8, ZmSFP10, ZmSTP15 and ZmSUT1 were down-regulated, while ZmSFP7 was up-regulated, during both embryo and endosperm development (Fig. 8). In particular, the expression levels of most tested ZmST genes, such as ZmINT4, ZmpGlcT4, ZmPMT4, ZmPMT5, ZmPMT8, ZmPMT9, ZmPMT13, ZmSFP5, ZmSFP7, ZmSFP9, ZmSFP10, ZmSTP7, ZmSTP15, ZmVGT1 and ZmSUT2, increased gradually in the early stage of grain filling, reaching a peak at 8-12 days after pollination, and then declined gradually (Fig. 9A). As supported by previous studies emphasizing the significance of the grain filling stage for seed quality and yield (Ji et al., 2022), we investigated the grain-filling rate of B73 over four days from 4 DAP to 28 DAP. Our findings revealed a substantial increase in the grain-filling rate during the developmental stages of 8-12 DAP and 16-20 DAP, aligning with the observed expression pattern of seed maturation (Fig. 9B). These results underlined the crucial role of ZmST genes in the embryonic, endospermic, and seed development stages.

Figure 8 The expression profiles of the ZmST genes in embryo and endosperm.

The qRT-PCR analysis was used to analyze the expression of selected ZmST genes in embryo (Em, shown in dark gray) and endosperm (En, shown in light gray). The names of the genes were labeled at the top of each diagram. 12D, 14D, 16D, 18D, 20D, 22D, 24D, 28D: embryo and endosperm of 12 DAP, 14 DAP, 16 DAP, 18 DAP, 20 DAP, 22 DAP, 24 DAP, 28 DAP, respectively. DAP, days after pollination. Columns were the mean of three independent replicates, and error bars represented SD. * and ** indicated significant differences with P < 0.05 and P < 0.01, respectively.

Figure 9 Expression profiles and grain-filling rate of the ZmST genes in seed.

(A) The qRT-PCR was used to analyze the expression levels of selected ZmST genes in seed (Se). The names of the genes were labeled at the top of each diagram. Se2, Se4, Se6, Se8, Se10, Se12, Se14, Se16, Se18, Se20, Se22, Se24, Se28: seed of 2 DAP, 4 DAP, 6 DAP, 8 DAP, 10 DAP, 12 DAP, 14 DAP, 16 DAP, 18 DAP, 20 DAP, 22 DAP, 24 DAP, 28 DAP, respectively. DAP, days after pollination. Columns were the mean of three independent replicates, and error bars represented SD. * and ** indicated significant differences with P < 0.05 and P < 0.01, respectively. (B) The grain-filling rate calculated by the weight increase of hundred-grain weight was measured from 4 to 28 DAP every 4 days.

Discussion

The structures, phylogenetic relationship, and functional evolutions of sugar transporters have been extensively studied in various plant species (Julius et al., 2017; Zhu et al., 2021). However, knowledge of their possible roles and regulation processes among different classes of sugar transporters in maize is still limited. In this study, we identified 68 ZmST genes in the maize genome and analyzed their physicochemical properties (Table 1). As we know, there were sixty-two ST genes in Arabidopsis (Büttner, 2007), sixty-five ST genes in grape (Afoufa-Bastien et al., 2010), sixty-six ST genes in strawberry (Liu et al., 2020), sixty-nine ST genes in rice (Deng et al., 2019), seventy-three ST genes in apple (Wei et al., 2014), seventy-five ST genes in pear (Li et al., 2015). The number of ST genes in maize was similar to those found in other plant species, suggesting that the functions of ST genes were relatively conserved in functions and crucial for plant growth and development throughout different species. Additionally, the SWEET genes, as a newly characterized group of sugar transporters, were identified 20 genes in maize, 23, 22, 24, 21, 29 and 19 genes in sorghum, pearl millet, foxtail millet, rice, barley and Brachypodium, respectively (Vinodh Kumar et al., 2023). The number of maize SWEET genes was comparable to those observed in other plants. These findings indicated that sugar transporters gene family as well as the SWEET gene family in maize, didn’t participate in gene expansion, maintaining the number of genes unchanged.

The subcellular localization of sugar transporters is crucial to explore their potential functions in various biological processes. It is known that sugar transporters are typically localized in membrane system of cells, facilitating the transport of sugar from source leaves to sink tissues, such as developing seeds, providing essential carbon and energy sources for plant growth and development. The sugar content is higher in sink tissues, which requires the proton pumps on the membrane to counteract sugar concentration gradient. The AtSTs in Arabidopsis were primarily distributed in the cell membrane and tonoplast (Hedrich, Sauer & Neuhaus, 2015). The ZmSWEETs in maize are mainly localized in the plasma membrane, vacuolar membrane, and chloroplast thylakoid membrane (Vinodh Kumar et al., 2023). Consistent with this, ZmST proteins were also mainly located on the cell membrane (Table 1). Based on the phylogenetic relationships in maize, Arabidopsis and rice, ZmSTs were clustered into eight groups (Fig. 1), consistent with the previous evolutionary classification of STs from other species. ZmSTs showed a closer relationship with ST proteins from rice, which also belonged to graminaceous crop. This result indicated that the functions of ST genes were relatively conserved, especially in graminaceous crops.

In this study, we identified that 68 ZmST genes were located on 10 chromosomes in the maize genome (Fig. 2A). In the ZmST gene family, we only observed 15 gene pairs with segmental duplication (Fig. 2B, Table 3). Therefore, segmental duplication was probably the predominant driver for the expansion of ST gene family in maize. Several duplicated gene pairs, such as ZmpGlcT1 and ZmpGlcT2, ZmPMT7 and ZmPMT10, ZmSFP3 and ZmSFP4, ZmSTP5 and ZmSTP20, ZmSUT1 and ZmSUT7, exhibited similar exon-intron structures and conserved motifs, indicating a certain degree of functional redundancy (Figs. 3B–3D). During the course of evolution, changes in gene structures may lead to variations in expression patterns and functions among ZmST genes.

Sugar transporters, divided into MFS-type sugar transporters and SWEET family, are critical for the proper functioning and coordination of various physiological processes during plants growth and development. Previous studies have highlighted the significant roles of MFS-type sugar transporters in different species. MdERDL6 and OsTST1 gene play important roles in plant development and kernel maturation in tomato and rice (Zhu et al., 2021; Yang et al., 2023). ZmSWEETs showed increased expression levels during maize seed germination, indicating their involvement in nutrient supply to the embryo axis (López-Coria et al., 2019). Moreover, ZmSWEET family members exhibited specific expression patterns in embryo and surrounding regions and upregulated expression under abiotic stress (Shen et al., 2022; Vinodh Kumar et al., 2023). So far, there have been many studies reported on sugar transporter proteins in corn, but they are mainly about the SWEET family proteins. However, there are very few reports about the MFS type in maize. Therefore, it is imperative to conduct extensive research on the MFS superfamily in maize to bridge this knowledge gap and gain a comprehensive understanding of their potential roles in maize kernel development. Gene expression patterns can help us understand the biological functions of genes. In Arabidopsis, the gene AtPLT5 was primarily expressed in sink tissues (Reinders, Panshyshyn & Ward, 2005), while in maize, a few members of ZmPMT group were expressed in seeds and vegetative organs like old leaves, roots and stems, indicating their involvement in sugar transport to specific sink tissues in plants. Neither AtSTPs in Arabidopsis nor ZmSTPs in maize were found in the female gametophyte or developing seeds (Büttner, 2010), suggesting that the STP family may not be involved in the seed maturation process. This suggestion was supported indirectly by previous research, demonstrating the involvement of ZmSTP2 and ZmSTP20 in maize disease resistance (Ma et al., 2022). AtERDL6 orthologs showed higher expression levels in fleshy fruits that accumulated a large amount of sugar, such as tomatoes (McCurdy et al., 2010), oranges (Zheng et al., 2014) and apples (Li et al., 2016). However, ZmSFPs in maize showed lower expression levels than fruits, indicating that the role of SFPs may not be prominent in crops. AtSUC2 was expressed in source-leaf companion cells of phloem (Stadler & Sauer, 1996), while OsSUT1 in rice played a role in sucrose transport during grain filling (Scofield et al., 2002). Similarly, certain ZmSUTs showed high expression levels in leaves and developing seeds, implying conserved functions of SUTs across Arabidopsis, rice and maize. It was reported that the zmsut2 mutant plant displayed slower growth, smaller ears, and reduced yield compared to WT (Leach et al., 2017). This result implied that the other six ZmSUTs may also play a similar role in seed development. AtTMT1 gene was strongly expressed in young developing tissues (Wormit et al., 2006), and ZmTSTs in maize exhibited higher expression levels in the young leaves, suggesting their involvement in sugar transport during rapid tissue expansion of cells in young tissues. The expression levels of OsTMT1 and OsTMT2 in rice were high in various organs excluding the endosperm (Cho et al., 2010), and similarly, ZmTSTs in maize were highly expressed in leaves, roots, stems, tassels, embryos and seeds, with lower expression in endosperm. Moreover, ZmTSTs showed higher expression levels in the early stage of grain filling compared to the late stage of grain filling, indicating their importance in early seed development. AtVGT1 was detected in all developmental stages and organs except roots in Arabidopsis (Aluri & Büttner, 2007), while ZmVGT genes were expressed in roots. Overall, most ZmPMT, ZmSFP and ZmSTP genes were predominantly expressed in vegetative organs and barely expressed in developing kernels, whereas, most ZmpGlcT, ZmVGT and ZmTST genes were highly expressed in seeds (Figs. 7, 8 and 9). Notably, most tested ZmST genes exhibited an increase in expression during the early stage of grain filling and a subsequent decline during seed development in maize, suggesting their potential role in grain filling and seed maturation. Overall, our findings deepen our understanding of the pivotal roles of these transporters during maize kernels development and maturation, with the potential to contribute to improving maize yield.

Conclusions

In summary, sixty-eight ZmST genes were identified and systematically analyzed in maize. Gene family analyses were conducted to investigate their physicochemical properties, chromosomal localizations, gene structures and biological functions in maize. The expression pattern analyses suggested that these ZmST genes may play a vital role in maize kernel development and could be potential candidates for improving maize yield. These significant findings will serve as valuable references for further research on ZmSTs and provide new genetic resources for high-yield maize breeding.

Supplemental Information

Figure S1 Predicted 3D structures of 68 ZmST proteins

Click here for additional data file.

Table S1 Primer sequences used for qRT-PCR in this study

Click here for additional data file.

Table S2 Amino acid sequences of ZmST proteins in maize

Click here for additional data file.

Table S3 Fifteen conserved motifs identified by MEME online tool

Click here for additional data file.

Supplemental Information 1 MIQE checklist

Click here for additional data file.

Dataset S1 The raw data for qRT-PCR experiments and the method of calculation

Click here for additional data file.

The authors thank the Maize Genetics and Genomics Database (https://maizegdb.org/) for the raw data.

Additional Information and Declarations

Competing Interests

Author Contributions

Data Availability

Nan Sun, Yanfeng Liu, Hongxia Zhang and Limin Wang, staff at Ludong University, completed the field experiment at Zhaoyuan Shenghui Agricultural Technology Development Co., Ltd. Zhaoyuan Shenghui Agricultural Technology Development Co., Ltd. has established a University-Industry-Research Cooperation with Ludong University.

Nan Sun conceived and designed the experiments, analyzed the data, prepared figures and/or tables, and approved the final draft.

Yanfeng Liu conceived and designed the experiments, analyzed the data, prepared figures and/or tables, and approved the final draft.

Tao Xu performed the experiments, prepared figures and/or tables, and approved the final draft.

Xiaoyan Zhou performed the experiments, prepared figures and/or tables, and approved the final draft.

Heyang Xu performed the experiments, prepared figures and/or tables, and approved the final draft.

Hongxia Zhang conceived and designed the experiments, analyzed the data, authored or reviewed drafts of the article, and approved the final draft.

Renhui Zhan performed the experiments, authored or reviewed drafts of the article, and approved the final draft.

Limin Wang conceived and designed the experiments, analyzed the data, authored or reviewed drafts of the article, and approved the final draft.

The following information was supplied regarding data availability:

The raw data are available in the Supplementary Files.

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
