# Peer review of "Genome-wide analysis of sugar transporter genes in maize (Zea mays L.): identification, characterization and their expression profiles during kernel development"

_PeerJ, doi:10.7717/peerj.16423_

## Round 0.1 · original submission · Major Revisions

Dear authors,

The manuscript entitled “Genome-wide analysis of sugar transporter genes in maize (Zea mays L.): identification, characterization and their expression profiles during kernel development” has been revised by two reviewers. The first reviewer recommended minor revisions while the second recommended rejecting the manuscript.

The authors in this study performed a comprehensive search in the maize genome and identified 68 ZmST genes. The authors applied phylogenetic relationship, chromosome location, collinearity analysis, conservative structures, and expression patterns analyses. The study detected that ZmSTs have a substantial role in sugar transportation and seed development. These findings provide fairly robust information for further research on ZmSTs and enhance maize breeding.

Accordingly, I recommend major revisions and give the authors an opportunity to improve their manuscript to be published in PeeJ.

Kindest regards
Elsayed Mansour

·

Basic reporting

The manuscript is well written and interesting but still needs some minor changes.

Experimental design

The study demonstrates a good experimental design.

Validity of the findings

No comments

Additional comments

I reviewed the paper titled "Genome-wide analysis of sugar transporter genes in maize (Zea mays L.): identification, characterization and their expression profiles during kernel development". The authors performed a comprehensive search against the updated maize genome B73_RefGen_v5 and identified 68 ZmST genes. Through phylogenetic relationship, chromosome location, collinearity analysis, conservative structures and expression patterns analyses in maize. They found that ZmSTs play a significant role in sugar transportation and seed development. They also reported that their results will serve as valuable references for further research on ZmSTs and provide new genetic resources for the high-yield maize breeding.
-Comments and Suggestions for Authors
Abstract
The abstract is well written.
Introduction
-The introduction section is comprehensive and well written.
-Please find more corrections as track changes in the manuscript pdf file.
Materials and methods
- In the expression analysis by qRT-PCR, if possible, I suggest the authors to mention the size of the amplicons in Supplemental_Table_S2 and from either the 5' end or the 3' end the amplicons were amplified
-Please find more corrections as track changes in the manuscript pdf file.
Results
-The results section is well written.
- Please find more corrections as track changes in the manuscript pdf file.
Discussion
-The discussion section is well written.
Conclusion
-The conclusion section is well written.
References
Please unify the style according to the journal instructions
Figure 3: Please add more information such as " Bootstrap values were calculated from 1000 replications and only the values with ?????? % bootstrapping were considered significant, and are indicated on the branch nodes. "

Reviewer 2 ·

Basic reporting

This manuscript identified the sugar transporters genes in maize and did the bioinformatics analysis. However, the experiments on the expression and function of these sugar transporters genes were lacking. It was not suitable for publication at its present status.

Experimental design

The authors did not analyze the SWEET transporter family, the title on sugar transporter genes is not suitable.
The research on the identification and function of MST and SUT in maize were not mentioned.

Validity of the findings

no comment

---

## Round 0.2 · Minor Revisions

The authors have adequately addressed all of the comments made by reviewer 1 in the revised version of the manuscript. In addition, scientifically responded to the criticisms of Reviewer 2. But still, the authors need to discuss how is this study different from the following previously published studies:

López-Coria M, Sánchez-Sánchez T, Martínez-Marcelo VH, Aguilera-Alvarado GP, Flores-Barrera M, King-Díaz B, Sánchez-Nieto S. SWEET Transporters for the Nourishment of Embryonic Tissues during Maize Germination. Genes (Basel). 2019 Oct 7;10(10):780. doi: 10.3390/genes10100780.

Yu-xin et al., 2022.
https://www.sciencedirect.com/science/article/pii/S2095311922003008?via%3Dihub

Shen et al., 2022. A transcriptional landscape underlying sugar import for grain set in maize. The Plant Journal, doi: 10.1111/tpj.15668

Vinodh Kumar PN, Mallikarjuna MG, Jha SK, Mahato A, Lal SK, K R Y, Lohithaswa HC, Chinnusamy V. Unravelling structural, functional, evolutionary and genetic basis of SWEET transporters regulating abiotic stress tolerance in maize. Int J Biol Macromol. 2023 Feb 28;229:539-560. doi: 10.1016/j.ijbiomac.2022.12.326.

There appear to be more. A more exhaustive literature search is needed with an explanation as to how these data in the current study are different and improve the knowledge of maize sugar transporters.

·

Basic reporting

The manuscript is well revised

Experimental design

The study demonstrates a good experimental design

Validity of the findings

No comments

Additional comments

The authors have amended the manuscript based on my previous comments

---

## Round 0.3 · accepted · Accept

The authors have adequately addressed all reviewers' comments. In addition, The authors have addressed the Editor's comment. Accordingly, the manuscript is ready for publication.